# How to Use Biomechanical Job Exposure Matrices with Job History to Access Work Exposure for Musculoskeletal Disorders? Application of Mathematical Modeling in Severe Knee Pain in the Constances Cohort

**DOI:** 10.3390/ijerph192316217

**Published:** 2022-12-04

**Authors:** Guillaume Deltreil, Patrick Tardivel, Piotr Graczyk, Mikael Escobar-Bach, Alexis Descatha

**Affiliations:** 1Univ. Angers, CHU Angers, Univ. Rennes, Inserm, EHESP, Irset (Institut de Recherche en Santé, Environnement et Travail)—UMRS 1085, SFR ICAT, 49100 Angers, France; 2CNRS, LAREMA, SFR MATHSTIC, Université d’Angers, 49100 Angers, France; 3UMR 5584 CNRS, Université de Bourgogne Franche-Comté, 21078 Dijon, France; 4Epidemiology and Prevention, Donald and Barbara Zucker School of Medicine, Hofstra University Northwell Health, Hempstead, NY 11549, USA; 5CHU Angers, Centre Antipoison—Centre de Données Cliniques, 49000 Angers, France

**Keywords:** occupational, musculoskeletal, pain, lifecourse, mathematical modeling

## Abstract

Introduction: Musculoskeletal disorders related to work might be caused by the cumulative effect of occupational exposures during working life. We aimed to develop a new model which allows to compare the accuracy of duration of work and intensity/frequency associations in application to severe knee pain. Methods: From the CONSTANCES cohort, 62,620 subjects who were working at inclusion and coded were included in the study. The biomechanical job exposure matrix “JEM Constances” was used to assess the intensity/frequency of heavy lifting and kneeling/squatting at work together with work history to characterize the association between occupational exposure and severe knee pain. An innovative model G was developed and evaluated, allowing to compare the accuracy of duration of work and intensity/frequency associations. Results: The mean age was 49 years at inception with 46 percent of women. The G model developed was slightly better than regular models. Among the men subgroup, odds ratios of the highest quartile for the duration and low intensity were not significant for both exposures, whereas intensity/duration were for every duration. Results in women were less interpretable. Conclusions: Though higher duration increased strength of association with severe knee pain, intensity/frequency were important predictors among men. Exposure estimation along working history should have emphasis on such parameters, though other outcomes should be studied and have a focus on women.

## 1. Introduction

Musculoskeletal disorders (MSDs) related to repetitive and physically demanding working conditions continue to represent one of the largest occupational disease in industrialized countries [1,2]. MSD related to work are caused by non-traumatic injuries, with a possible cumulative effect of occupational exposures during working life, mainly for degenerative disorders, such as osteoarthritis, and severe knee pain [3]. Actually, knee pain was considered because it is a very common pain in occupational setting with clear work exposure [4].

The evaluation of biomechanical exposures can be done in different ways and expressed around two main dimensions, intensity/frequency per day and duration over the year [5]. The exposure assessment can be obtained by estimations based on subjective judgments (self-reports, expert judgments), systematic observations (observations at the workplace, video recording), and direct measurements (at the workplace or in laboratory). However, these techniques are problematic when past exposure evaluations are needed.

Job exposure matrices (JEMs) are commonly used in occupational epidemiology research for the evaluation of past exposures [6]. Indeed, JEMs allow estimating participants exposures to occupational factors based on job titles, industry sector, and population exposure data. Several biomechanical JEMs have become available recently [7,8,9,10,11]. A cumulative exposure index is commonly used to assess cumulative work exposure by multiplying duration and intensity/frequency. However, it is not clear how to consider the combination of intensity/frequency when assessing exposure over the years.

Thus, before optimizing models using relevant statistical methods, we first aimed to determine if low level exposure with high duration is equivalent to high level exposure with low duration in the example of severe knee pain and two occupational exposures: carrying heavy loads and kneeling/squatting.

We then aimed develop a new model and compare the accuracy of duration of work and intensity/frequency associations in application to the knee disorders using severe knee pain as an outcome in a large cohort study, by developing an innovative model.

## 2. Materials and Methods

### 2.1. Population

The CONSTANCES study is a French general population-based cohort [12]. More than 200,000 participants, aged 18–69 years, were recruited between 2012 and 2020 in 23 health screening centers across France. The recruitment was limited to people affiliated to the French National Health Insurance Fund who correspond to active or former salaried workers and their families and excludes agricultural and self-employed workers [12]. At enrolment, self-administered questionnaires were sent to participants to collect data, including lifestyle, life events, health, and occupations. Variables of interest were collected from the baseline self-administered questionnaire and the medical interview.

For this work, we used French CONSTANCES clean data extracted in 2020. Subjects from this cohort were active at their inclusion with work trajectory coded.

### 2.2. Variables of Interest

Participants’ sex, age at inception, known inflammatory disease of the joints, regular leisure activity (sports, gardening yes/no), Center for Epidemiologic Studies-Depression Scale (CES-D) into two categories (yes/no), were retrieved from the baseline questionnaire, and body mass index from the medical examination.

JEM Constances, which is based on self-reported exposure was used to evaluate occupational exposure [13]. In the JEM, occupational exposure is rated from 0 to 4 for intensity/frequency of heavy lifting (“lifting”) and 1 to 4 of kneeling of squatting (“kneeling”) based on reported job titles. The JEM Constances was combined with participants reported work trajectory that were coded at baseline retrospectively.

The main outcome was reporting severe knee pain, collected from the self-reported questionnaire at inception: yes if knee pain intensity >5/10 or having knee pain for more than a month per year.

### 2.3. Mathematical Modeling

In order to study the influence of duration and intensity/frequency on the onset of the disease, several logistic regression models were built for the two exposures separately: heavy lifting and kneeling.

The method we propose in this article is based on a generalization of the logistic regression approach. Formally, we define ${\left(Y,X_{i1},\cdots ,X_{ip}\right)}_{\left[1,n\right]}$ an i.i.d. sample in $\left\{0,\;1\right\}\times R^{p}$ of size $n\in N^{*}$ where *Y* is the response variable and corresponds to the illness status of each subject (if the subject is sick Y=1, else Y=0) and $X_{i}=\left(X_{i1},\cdots ,X_{ip}\right)$ are the variables $X_{i1},\cdots ,X_{ip-6}$ corresponding to a score based on occupation times and levels exposures and $X_{ip-5},\cdots ,X_{ip}$ corresponding to others adjustment variables.

For a given *i* conditionally on $X_{i}:=\left(X_{i1},...,X_{ip}\right)$, $Y_{i}$ follows a Bernoulli distribution, such as: (1)$$P\left(Y_{i}=1|X_{i}\right)=\pi_{\beta^{*}}\left(X_{i}\right):=\frac{\exp(<\beta^{*},X_{i}>)}{1+\exp(<\beta^{*},X_{i}>)},$$
where $\beta^{*}=\left(\beta_{0}^{*},\beta_{1}^{*},\cdots ,\beta_{p}^{*}\right)\in R^{p}$ is an unknown vector of parameters to be estimated. We estimate the parameter $\beta^{*}$, given in (Equation 1), by Maximal Likelihood Estimation, i.e., by minimizing the normalized opposite of the log likelihood $\gamma_{n}\left(\beta \right)$ over $R^{p}$: $$\hat{\beta }=arg\min_{\beta \in R^{p}}\gamma_{n}\left(\beta \right).$$

#### 2.3.1. Statistical Models

For the occupational health data from the cohort CONSTANCES, several logistic models were possible, depending on the total duration value of the careers and the average exposures of the individuals. We describe them in our cohort via the variables $T_{i}=\left(T_{i1},\cdots ,T_{ik_{i}}\right)$, $N_{i}=\left(N_{i1},\cdots ,N_{ik_{i}}\right)$, and $a_{1i},\cdots ,a_{6i}$, respectively, for occupation times, levels of exposures for each $k_{i}$ jobs held by the subject *i*, as well as six adjustment variables ($a_{1i}$ = sex, $a_{2i}$ = age, $a_{3i}$ = imc, $a_{4i}$ = leisure, $a_{5i}$ = arthrite, $a_{6i}$ = depression). We must take into account that the number of jobs $k_{i}$ can be very different. The models *A*, *B*, and *C* are defined through the three following transformations:$$\begin{array}{l} X_{i}^{A}=\sum_{j=1}^{k_{i}}N_{ij}T_{ij}\\ X_{i}^{B}=\sum_{j=1}^{k_{i}}N_{ij}^{B}T_{ij},\;\mathrm{with}N_{ij}^{B}=\left\{\begin{array}{cc} 0, & \mathrm{if}N_{\mathrm{ij}}\;\leq \;1\\ N_{\mathrm{ij}}, & \mathrm{if}\;\mathrm{else}. \end{array}\right.\\ X_{i}^{C}=\sum_{j=1}^{k_{i}}N_{ij}^{C}T_{ij},\;\mathrm{with}N_{ij}^{C}=\left\{\begin{array}{cc} 0, & \mathrm{if}N_{\mathrm{ij}}\;\leq \;3\\ 1, & \mathrm{if}\;\mathrm{else}. \end{array}\right. \end{array}$$


Thus, $Y_{i}$ follows a Bernoulli distribution, such that $P\left(Y_{i}=1|X_{i}\right)=\mathrm{logit}\left(\beta_{0}^{*}+\beta_{1}^{*}X_{i}^{•}+\sum_{j=1}^{6}\beta_{j+1}^{*}a_{ji}\right)$ where $X_{i}^{•}$ is to be replaced by $X_{i}^{A}$, $X_{i}^{B}$ or $X_{i}^{C}$ depending on the variable considered. We specify that the model *B* differs, from the model *A*, because in the computation of its transformation the exposure levels between 0 and 1 are confounded. The model *C* differs from the model *A*, because in the computation of its transformation the lowest exposure levels are nullified. Actually, models *B* and *C* are byproducts of the classical model *A*, standard in the literature dealing with cumulative exposure, such as smoking habits or cumulative exposure index at work [14]. More precisely, in Model *B*, in the sum of products “duration by exposure”, we neglect small exposures by thresholding to zero exposures smaller or equal to 1 [15]. In Model *C*, we just consider the total duration restricted to jobs physically very demanding (exposure larger than 3) [16]. In order to define the model *G* we need to introduce two transformations $\phi_{1}$ and $\phi_{2}$:$\phi_{1}\left(T_{i}\right)=\sum_{j=1}^{k_{i}}T_{ij}$ corresponding to the total duration of the career range.$\phi_{2}\left(T_{i},N_{i}\right)=\frac{\sum_{j=1}^{k_{i}}N_{ij}g\left(T_{ij}\right)}{\sum_{j=1}^{k_{i}}g\left(T_{ij}\right)}$, where g(x)=1−exp(−x) which is the time-weighted average of the exposure level. This weighting was considered to emphasise intensity, especially for short duration exposure.

The construction of the design matrix ${\left(X_{ij}\right)}_{\left(1\leq i\leq n,1\leq j\leq p\right)}$ of the model *G* is based on a semi-coding of the variables ${\left\{\phi_{1}\left(T_{i}\right)\right\}}_{1\leq i\leq n}$ and ${\left\{\phi_{2}\left(T_{i},N_{i}\right)\right\}}_{1\leq i\leq n}$. More precisely, the membership in a group is determined by the belonging of the values $\phi_{1}\left(T_{i}\right)$ and $\phi_{2}\left(T_{i},N_{i}\right)$ to different given intervals. For this study, we consider the classes of intervals
$$G_{1}:=\left\{\left[q_{j},q_{j+1}\right],\;j=1,\ldots ,q\right\}\;\mathrm{and}\;G_{2,c}:={\left\{\;\left[j-1,j\right]\;\right\}}_{c\leq j\leq 4}\cup \left\{4\right\},c=1,2$$
where $q_{j}\in N^{*}$ defines fixed empirical quantiles of the sample ${\left\{\phi_{1}\left(T_{i}\right)\right\}}_{1\leq i\leq n}$. When c=1, the exposure considered is carrying a heavy load, and when c=2 the exposure considered is kneeling. The set of groups is defined as the class $G:=\left\{G_{j},\;j=1,\ldots ,p\right\}=G_{1}\times G_{2}$ where p=(6−c)q. An individual is then associated with the group j∈[[1,p]] with $X_{ij}=1$ if and only if $\left(\phi_{1}\left(T_{i}\right),\phi_{2}\left(T_{i},N_{i}\right)\right)\in G_{j}.$

#### 2.3.2. Selection of the Design Matrix for Model *G*

We propose to choose the design matrix *X* for the model *G* by a recent model selection procedure, introduced and described in the article “Model selection in logistic regression” by Kwemou et al. [17]. The mathematical guarantees for this model selection method are based on oracle inequalities from Birgé and Massart [18]. This model selection is performed using penalized maximum likelihood estimators which will allow us to choose the best design matrix.

Let F be a family of design matrices $X^{\left(m\right)}$, m∈{1,⋯,M}. For each m∈{1,⋯,M}, we define ${\hat{\beta }}_{m}$ as the estimator obtained by minimizing $\gamma_{n}$ over $R^{p_{m}}$ where $p_{m}$ is the number of columns of $X^{\left(m\right)}$, namely: $${\hat{\beta }}_{m}=\arg\min_{\beta \in R^{p_{m}}}\gamma_{n}\left(\beta \right).$$
We use a data driven strategy that selects the best matrix among the family F. For this purpose we use a penalized maximum likelihood criterion for choosing the index *m* associated to the appropriate design. Here, we consider the Akaike information criterion where $p_{m}$ corresponds to the number of parameters in the model (Equation 1), i.e., $p_{m}$ is the number of columns of the matrix $X^{\left(m\right)}$ and
$$\hat{m}=\arg\min_{m\in \{1,\cdots ,M\}}\left\{\gamma_{n}\left({\hat{\beta }}_{m}\right)+p_{m}\right\}.$$
Hence, minimizing this criterion allows us to find the best design matrix $X^{\hat{m}}$.

The “lifting” and “kneeling” quartiles of high exposure are considered in terms of duration and intensity/frequency:into 5 categories for heavy lifting $G_{2,1}:={\left\{\;\left[j-1,j\right]\;\right\}}_{1\leq j\leq 4}\cup \left\{4\right\}$,into 4 categories for kneeling $G_{2,2}:={\left\{\;\left[j-1,j\right]\;\right\}}_{2\leq j\leq 4}\cup \left\{4\right\}$.

### 2.4. α-Divergence and Statistical Tests

The logistic regression formula given in (Equation 1) depends on the model A,B,C, and *G*. We wish to compare these statistical models, via a differentiation criterion, in order to identify the most appropriate model for this formula. Let *r* be the unknown probability that a randomly selected subject from the population is sick. It is natural to estimate the unknown parameter *r* by the proportion of patients observed in our dataset: ${\hat{r}}_{1}=\frac{1}{n}\sum_{i=1}^{n}Y_{i}$. One can also estimate *r* from the logistic regression formula via the estimator ${\hat{r}}_{2}=\frac{1}{n}\sum_{i=1}^{n}\pi_{\beta^{*}}\left(X_{i}\right)$. Note that unlike ${\hat{r}}_{1}$, the estimator ${\hat{r}}_{2}$ depends on the model. For an appropriate model it is natural to choose the model for which the estimators ${\hat{r}}_{1}$ and ${\hat{r}}_{2}$ are close.

We here chose to rely on the work of A. Basu et al. [19] who developed a robust estimator for the density function. The criterion used in this article is based on the α-divergence between two densities *f* and *g* (relative to a measure μ) defined for an α>0 as follows: $$d_{\alpha }\left(g,f\right)=\int \left\{f^{1+\alpha }\left(z\right)-\left(1+\frac{1}{\alpha }\right)g\left(z\right)f^{\alpha }\left(z\right)+\frac{1}{\alpha }g^{1+\alpha }\left(z\right)\right\}d\mu \left(z\right).$$
Here, we need to use the α-divergence between two Bernoulli laws with parameters $r_{1}\in \left[0,1\right]$ and $r_{2}\in \left[0,1\right]$ which we define by
$$\begin{array}{c} d_{\alpha }\left(r_{1},r_{2}\right)=\left\{\begin{array}{cc} r_{2}^{1+\alpha }+{\left(1-r_{2}\right)}^{1+\alpha }-\left(1+\frac{1}{\alpha }\right)\left({\left(1-r_{2}\right)}^{\alpha }\left(1-r_{1}\right)+r_{2}^{\alpha }r_{1}\right)\\ +\frac{1}{\alpha }\left({\left(1-r_{1}\right)}^{1+\alpha }+r_{1}^{1+\alpha }\right) & \alpha >0\\ r_{1}\ln\left(\frac{r_{1}}{r_{2}}\right)+\left(1-r_{2}\right)\ln\left(\frac{1-r_{1}}{1-r_{2}}\right) & \alpha =0 \end{array}\right. \end{array}$$
The parameter α here defines a trade-off between the estimation efficiency and the variability robustness. Note that when α is close to 0 we retrieve the Kullback–Leibler divergence and when α=1 we have $d_{1}\left(r_{1},r_{2}\right)=2{\left(r_{1}-r_{2}\right)}^{2}$.

Next, we compared the *p*-value of the Wald’s test (with Bonferroni correction) and odds ratios (OR) of the highest quartile for the duration with low intensity/frequency and the highest quartile for the intensity/frequency with low duration. Stratification on sex was performed as sensitivity analysis. For both exposures, logistic models were built adjusted on relevant variables. We compared the *p*-value of the Wald’s test (with Bonferroni adjustments) and odds ratios of different quartile of duration and intensity/frequency and with low duration and lowest intensity as reference.

### 2.5. Analysis Plan

After a brief description of the sample and the available adjustment variables, we assessed the performance score based on the α-divergence allowing the comparison of different models with α varying with each relevant occupational exposure. For a small value of parameter α, the decision of the model choice can be questioned but when α increases the choice of model becomes easier and the model G is selected.

Then, we were able to compare Odds ratios with the Wald’s test. Sex stratification has been performed as primary analysis (whereas 45 years of age stratification has been considered in secondary analysis).

Analyses were performed on Python 3.8.8 (statsmodels, statistics, pandas, numpy).

## 3. Results

The sample included 62,620 subjects (Table 1), with 29,064 women (46.4%) and with a median of 48.1 years old (20 years of employment). The subjects reported regular primary leisure activities in 43.3% of cases, with half of sample who were overweight/obese, and an important part (22.5%). who had a positive CES-D, suggesting a depression.

In order to study intensity and duration of exposure, we first selected the best model between *A* to *G*. We compared this family of models by increasing α. For any value of parameter alpha, the model *G* is selected (Table 2 and Table 3).

For heavy lifting as well as kneeling, intensity/frequency was the most important predictor. Duration increased risks only for men with a dose–response relationship. For women, only intensity and frequency in low duration for both exposures seemed associated with the knee pain (Table 4 and Table 5). Although the statistical power was lower, the pattern of association was similar for participants aged less and more than 45 years (Table 6 and Table 7).

## 4. Discussion

This is the first study that used a developed mathematical model to compare the effect of duration and intensity during working life, on the association with severe knee as a proxy of degenerative musculoskeletal disorders. The new model *G* was found to be better than the usual ones, though the difference was minor. For men, we found that the OR of the highest quartile for the duration and low intensity is not significant for both exposures, whereas intensity/duration is significant for every duration, with a dose–response relationship. Results for women were limited. As expected, there was as an important effect of the intensity of heavy lifting and kneeling on severe knee pain. Both exposures are known to be associated with knee disorders [20,21,22]. The dose–response relationship has been described previously. Jensen [23,24] calculated an equivalent of our model A using an individual exposure from the number of knee-straining activities and the number of years in the trade within a collective of floor layers, carpenters, and compositors. The ORs for knee complaints and radiographically determined knee osteoarthritis were 3.0 (95% CI, 0.5 to 17.2) in the low-exposure group, 4.2 (95% CI, 0.6 to 27.6) in the medium-exposure group, and 4.9 (95% CI, 1.1 to 21.9) in the high-exposure group compared with the zero-exposure group [23]. There is an important difference in the strength of the associations compared to our work but it should be explained by the large population design with JEM exposure methods. Indeed, high exposure is considered using the proxy of job title with a large variation inside job categories.

Similarly expected, in a previous review on occupational exposure and knee osteoarthritis, ref. [24] lifting and carrying of loads was significantly associated with severe knee pain. Knee osteoarthritis was also found associated with lifting and carrying of loads with a dose–response relationship: OR of 2.0 (95% CI, 1.1 to 3.6) in the exposure group 630 to <5120 kg-hours over life, up to an OR of 2.6 (95% CI, 1.1 to 6.1) in the highest exposure group (>37,000 kg-hours over life) in men [23]. We included in our study the category “Inflammatory osteoarthritis” even if its frequency was low (1.4%). Indeed, such disorders are known to be associated with knee pain [25]. In our study, the proportion of subjects with such diseases suffering from knee pain is high: 31.0% whereas, in the entire population, the proportion of subjects having knee pain is only 13.6%; when testing whether the regression coefficient of the covariate “Inflammatory osteoarthritis” is null we obtain a *p*-value, with the Wald’s test, smaller than 10${}^{-9}$ with 95% confident for the odds ratio is [2.36, 3.19].

The lack of clear association for women was also found by D’Souza et al. who reported on an analysis of the US national survey, where they describe relationships between work activities and symptomatic knee osteoarthritis [26]. A significant exposure-response relationship was only found between symptomatic knee osteoarthritis and kneeling in men but not women. Different explanations might be suggested: since our model included adjusting factors, such as BMI and depression, there might be a more complex causal pathways than in men, such as considered in back pain [27]. JEM Constances is not gender stratified and applying a specific JEM for sex could be a focus for another study. Furthermore, a selection effect similar to healthy workers effect is also possible. The main strength of our study was the possibility to use JEM with working life course on a large cohort study. Limitations might also be raised by using a large but not representative population of French workers, and specific jobs in agriculture (not included by design) and mining (almost disappeared in France) should be considered since they are known factors related to lower limb MSD [22]. Second, the outcome was focused on severe knee pain. However, it is self-declared and might correspond to heterogenous disorders, work-related or not. It was used as a good example of a proxy of degenerative musculoskeletal disorders, and the use of pain intensity and severity is recommended [28]. We have previously shown that working in a kneeling or squatting position was significantly associated with severe knee pain [21]. More recently, similar trends of associations between severe knee pain and knee arthroplasty groups were showed in the same cohort [29]. This result is also found elsewhere, with non-managerial jobs associated with higher prevalence of knee osteoarthritis and knee symptoms [30]. Third, as we already mentioned, the use of JEMs might also be questioned since it is a global average evaluation that does not consider the differences inside job [31]. However, assessing exposure during long periods of time and for a big number of subjects is challenging and JEMs are appropriate tool to consider. Furthermore, assessment of carrying heavy loads exposure using JEMs was found to be valid compared to a self-administrated questionnaire [32]. Furthermore, biomechanical JEM used through the working life was already studied and even when the work environment have changed, application of a 4-scale at the individual level did not change regardless of period of time considered [33]. Fourth, activities at work can be varied and diverse, even if the gradations used can allow us to obtain a general idea of exposures. This could be more specific for each job and introduce some complementary variable to adjust the variability between mechanical exposure into different jobs. Fifth, since leisure activities can be numerous, the mechanical exposure coming from these activities and their intensity can train subjects and make them more resilient to disorders.

## 5. Conclusions

This innovative approach using mathematical modeling of working history and a JEM, shows that duration in years has a smaller impact than frequency/intensity and should be considered at least among men. Our new model *G* seems to be an interesting approach though the improvement is slight. Implications for potential policymakers and human resource management might been considered to achieve prevention of such pain during with the help of occupational practitioners. However, further studies should be completed on other outcomes, and have a focus on women.

## Figures and Tables

**Table 1 ijerph-19-16217-t001:** Description of the sample.

Variables		N (%)	Mean (SD)
Sex	Men	33,556 (53.6)	
	Women	29,064 (46.4)	
Age (years)			48.1 (13.1)
Body Mass Index (kg/m²)			25.02 (4.4)
Leisure activities	Yes	27,115 (43.3)	
	No	35,505 (56.7)	
Inflammatory Ostearthritis	Yes	875 (1.4)	
	No	61,745 (98.6)	
Depression	Yes	14,095 (22.5)	
	No	48,525 (77.5)	

**Table 2 ijerph-19-16217-t002:** Comparison of performance score for severe knee pain with lifting.

α	A	B	C	G
0	0.305	0.329	0.326	**0.192**
0.25	0.162	0.228	0.218	**0.019**
0.5	0.155	0.221	0.211	**0.017**
0.75	0.146	0.210	0.200	**0.015**
1	0.136	0.198	0.189	**0.013**

**Table 3 ijerph-19-16217-t003:** Comparison of performance score for severe knee pain with kneeling.

α	A	B	C	G
0	0.282	0.304	0.297	**0.194**
0.25	0.101	0.160	0.145	**0.021**
0.5	0.096	0.153	0.138	**0.018**
0.75	0.092	0.143	0.13	**0.016**
1	0.088	0.134	0.121	**0.014**

**Table 4 ijerph-19-16217-t004:** Results of adjusted logistic regression of severe knee pain with lifting, for men and women separately.

Variable		Men			Women		
Duration	Intensity/ Frequency	OR	IC 95%	*p*-Value	OR	IC 95%	*p*-Value
low	[1, 2)	1.83	[1.53, 2.19]	<$10^{-4}$	1.46	[1.29, 1.66]	<$10^{-4}$
low	[2, 3)	2.08	[1.71, 2.52]	<$10^{-4}$	1.47	[1.20, 1.81]	<$10^{-4}$
low	[3, 4)	1.96	[1.42, 2.70]	<$10^{-4}$	1.38	[0.91, 2.09]	0.13
low	{4}	2.49	[1.86, 3.33]	<$10^{-4}$	2.81	[1.60, 4.92]	<$10^{-4}$
medium	[0, 1)	1.07	[0.91, 1.27]	0.41	0.86	[0.77, 0.96]	0.01
medium	[1, 2)	1.63	[1.35, 1.97]	<$10^{-4}$	1.11	[0.97, 1.28]	0.13
medium	[2, 3)	2.39	[1.97, 2.90]	<$10^{-4}$	1.28	[1.02, 1.61]	0.04
medium	[3, 4)	2.45	[1.78, 3.39]	<$10^{-4}$	1.08	[0.62, 1.86]	0.79
medium	{4}	2.71	[1.84, 3.98]	<$10^{-4}$	1.09	[0.41, 2.93]	0.86
high	[0, 1)	1.09	[0.92, 1.29]	0.33	0.87	[0.77, 0.98]	0.03
high	[1, 2)	1.77	[1.49, 2.11]	<$10^{-4}$	1.07	[0.93, 1.23]	0.36
high	[2, 3)	2.12	[1.77, 2.53]	<$10^{-4}$	1.09	[0.86, 1.37]	0.49
high	[3, 4)	2.41	[1.87, 3.11]	<$10^{-4}$	1.37	[0.99, 1.91]	0.06
high	{4}	3.56	[2.68, 4.72]	<$10^{-4}$	1.67	[0.68, 4.12]	0.27

Adjusted on body mass index, leisure activity, inflammatory osteoarthritis, depression, and age/sex when not stratified; Reference duration low, Intensity/Frequency [0, 1).

**Table 5 ijerph-19-16217-t005:** Results of adjusted logistic regression of severe knee pain with kneeling, men and women separately.

Variable		Men			Women		
Duration	Intensity/ Frequency	OR	IC 95%	*p*-Value	OR	IC 95%	*p*-Value
low	[2, 3)	1.97	[1.67, 2.31]	<$10^{-4}$	1.34	[1.16, 1.53]	<$10^{-4}$
low	[3, 4)	1.75	[1.40, 2.20]	<$10^{-4}$	1.44	[1.22, 1.71]	<$10^{-4}$
low	{4}	1.98	[1.51, 2.60]	<$10^{-4}$	1.35	[1.01, 1.82]	0.04
medium	[1, 2)	1.06	[0.91, 1.23]	0.48	0.83	[0.74, 0.93]	<$10^{-4}$
medium	[2, 3)	1.86	[1.57, 2.20]	<$10^{-4}$	1.02	[0.88, 1.17]	0.83
medium	[3, 4)	2.19	[1.72, 2.79]	<$10^{-4}$	1.23	[1.03, 1.48]	0.02
medium	{4}	2.66	[1.91, 3.71]	<$10^{-4}$	0.91	[0.59, 1.39]	0.66
high	[1, 2)	1.10	[0.94, 1.28]	0.26	0.85	[0.76, 0.96]	0.01
high	[2, 3)	1.80	[1.54, 2.11]	<$10^{-4}$	1.03	[0.89, 1.19]	0.67
high	[3, 4)	2.16	[1.78, 2.62]	<$10^{-4}$	0.88	[0.74, 1.04]	0.13
high	{4}	2.55	[1.99, 3.28]	<$10^{-4}$	1.38	[1.03, 1.84]	0.03

Adjusted on body mass index, leisure activity, inflammatory osteoarthritis, depression, and age/sex when not stratified; Reference duration low, Intensity/Frequency [1, 2).

**Table 6 ijerph-19-16217-t006:** Results of adjusted logistic regression of severe knee pain with lifting, for <45 years old and 45 or more participants.

Variable		<45 Years			≥45 Years		
Duration	Intensity/ Frequency	OR	IC 95%	*p*-Value	OR	IC 95%	*p*-Value
low	[1, 2)	1.59	[1.35, 1.87]	<$10^{-4}$	1.51	[1.26, 1.81]	<$10^{-4}$
low	[2, 3)	1.71	[1.37, 2.14]	<$10^{-4}$	1.62	[1.31, 2.00]	<$10^{-4}$
low	[3, 4)	1.45	[0.95, 2.19]	0.08	1.42	[0.93, 2.16]	0.107
low	{4}	2.63	[1.83, 3.79]	<$10^{-4}$	1.31	[0.86, 2.02]	0.212
medium	[0, 1)	0.87	[0.75, 1.02]	0.09	0.92	[0.79, 1.06]	0.234
medium	[1, 2)	1.33	[1.10, 1.59]	0.002	1.24	[1.05, 1.46]	0.01
medium	[2, 3)	1.64	[1.28, 2.10]	<$10^{-4}$	1.57	[1.28, 1.93]	<$10^{-4}$
medium	[3, 4)	1.79	[1.19, 2.70]	0.005	1.95	[1.34, 2.84]	0.001
medium	{4}	3.30	[2.04, 5.34]	<$10^{-4}$	2.25	[1.42, 3.58]	0.001
high	[0, 1)	0.93	[0.77, 1.12]	0.442	0.88	[0.78, 1.01]	0.073
high	[1, 2)	1.31	[1.07, 1.62]	0.01	1.21	[1.05, 1.39]	0.007
high	[2, 3)	1.93	[1.51, 2.47]	<$10^{-4}$	1.46	[1.25, 1.71]	<$10^{-4}$
high	[3, 4)	1.36	[0.85, 2.19]	0.204	1.69	[1.36, 2.09]	<$10^{-4}$
high	{4}	1.63	[0.89, 2.98]	0.114	2.58	[1.97, 3.37]	<$10^{-4}$

Adjusted on body mass index, leisure activity, inflammatory osteoarthritis, depression, and age/sex when not stratified; Reference duration low, Intensity/Frequency [0, 1].

**Table 7 ijerph-19-16217-t007:** Results of adjusted logistic regression of severe knee pain with kneeling, for <45 years old and 45 or more participants.

Variable		<45 Years			≥45 Years		
Duration	Intensity/ Frequency	OR	IC 95%	*p*-Value	OR	IC 95%	*p*-Value
low	[2, 3)	1.50	[1.28, 1.78]	<$10^{-4}$	1.49	[1.24, 1.79]	<$10^{-4}$
low	[3, 4)	1.62	[1.31, 2.01]	<$10^{-4}$	1.45	[1.16, 1.80]	0.001
low	{4}	1.80	[1.33, 2.44]	<$10^{-4}$	1.40	[1.04, 1.88]	0.025
medium	[1, 2)	0.86	[0.74, 1.00]	0.06	0.91	[0.79, 1.05]	0.19
medium	[2, 3)	1.45	[1.22, 1.73]	<$10^{-4}$	1.20	[1.02, 1.41]	0.028
medium	[3, 4)	1.34	[1.04, 1.73]	0.026	1.47	[1.20, 2.47]	<$10^{-4}$
medium	{4}	1.24	[0.78, 1.97]	0.361	1.75	[1.24, 2.47]	0.001
high	[1, 2)	0.91	[0.76, 1.09]	0.300	0.88	[0.78, 1.00]	0.050
high	[2, 3)	1.38	[1.14, 1.69]	0.001	1.23	[1.08, 1.41]	0.002
high	[3, 4)	1.59	[1.22, 2.09]	0.001	1.22	[1.05, 1.42]	0.011
high	{4}	1.42	[0.89, 2.26]	0.139	1.73	[1.41, 2.12]	<$10^{-4}$

Adjusted on body mass index, leisure activity, inflammatory osteoarthritis, depression, and age/sex when not stratified; Reference duration low, Intensity/Frequency [1, 2).

## Data Availability

Personal health data underlying the findings of our study are not publicly available due to legal reasons related to data privacy protection. However, the data are available upon request to all interested researchers after authorization of the French “Commission nationale de l’informatique et des libertés”. The CONSTANCES email address is contact@constances.fr.

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
