# Peer review of "How to Use Biomechanical Job Exposure Matrices with Job History to Access Work Exposure for Musculoskeletal Disorders? Application of Mathematical Modeling in Severe Knee Pain in the Constances Cohort"

_ijerph, 2022, doi:10.3390/ijerph192316217_

Round 1

Reviewer 1 Report

This is a very interesting topic to evaluate work exposure for knee disorders using JEMs. The manuscript is very well written. But I have questions regarding the analysis itself.

(1) The major concern is whether the model comparison is fair. For model A, B and C, there is no weight for the total duration value. All models just multiply the total duration value by the average exposures. This does not account for the fact that some work and career can be longer and some can be shorter, and simply multiplying the total duration by the average exposure will give more weight for those work that last longer. In fact there can be a weight adjustment for Tij for model A, B and C. 

Also as each participant has different number of jobs, the proposed models do not consider much how to adjust the number of jobs in the analysis.

Also the authors said in line 104-105 there is a time-weighted average of the exposure level but there is no rationale explained how the g(x) was proposed.

(2) In line 54-55, it is said that the data is from 2020. Does this mean the data is a work trajectory for each participant in year 2020, or any past working history regardless which year it was? It is not clear. If it is a past working history, then also needs to consider the change in the working environment over the years and corresponding changes in the working environment could impact the work exposure.

(3) In Table 1, The percentage of 'Yes' for Inflammatory ostearthritis is just 1.4%. With such a low frequency in one category, I wonder if it is appropriate to be included as a covariate in a logistic regression model.

(4) Line 93-95: not sure how the cut-offs 1 and 3 were determined: if Nji<=1 then Nji(b)=0 or if Nji<=3 then Nji(c)=0. There was no explanation on these two proposed cut-offs.

(5) Line 174 said half of sample who were overweight/obese, but Table 1 Description of the sample does not contain source data to support this conclusion.

Author Response

Reviewer #1:

The major concern is whether the model comparison is fair. For model A, B and C, there is no weight for the total duration value. All models just multiply the total duration value by the average exposures. This does not account for the fact that some work and career can be longer and some can be shorter, and simply multiplying the total duration by the average exposure will give more weight for those work that last longer. In fact, there can be a weight adjustment for Tij for model A, B and C.

Also as each participant has different number of jobs, the proposed models do not consider much how to adjust the number of jobs in the analysis.

Also the authors said in line 104-105 there is a time-weighted average of the exposure level but there is no rationale explained how the g(x) was proposed.

R1.1. We thank the reviewer for his suggestions. Actually, as you pointed out in your comment, models A,B and C should take into account the number of jobs occupied by a subject. Unfortunately, there is a typo in formulas of these models. The letter (and not m) represents the number of jobs and thus these formulas should be (cf .pdf)

The formula A is a standard in the literature which deals with cumulative exposure, for example smoking habits or cumulative exposure index at work [1]. There may be a misunderstanding since model A is not the product of the total length by the average exposure. Actually, model A (whose formula is reminded above) is the sum of products “duration by exposure” and can be intuitively written as follows: duration first job x exposure first job+...+duration last job x exposure last job (the same as the pack.years method for estimating total smoking habit). Models B and C are byproducts of model A (see comment 4).

We agree that in the former version of the manuscript there is no rational explanation for the function g(x). The more natural way to compute a time weighted average exposure is to use, as a weight, “the fraction of the duration by the total duration”. However, we believe that a job with a short duration should be weighted more since a subject who occupy a new job can experiment an important stress.

The function is devoted to satisfying the above point. Indeed  and so when t is very small, . Consequently, when Tij is a very short duration we have (cf pdf)

Therefore, for a short duration, the weight obtained with the function g is larger than the standard weight  (cf .pdf).

We have added in the text: “This weighting was considered to emphasis intensity, especially for short duration exposure”.  

In line 54-55, it is said that the data is from 2020. Does this mean the data is a work trajectory for each participant in year 2020, or any past working history regardless which year it was? It is not clear. If it is a past working history, then also needs to consider the change in the working environment over the years and corresponding changes in the working environment could impact the work exposure.

R1.2. We have clarified the data was extracted in 2020. For biomechanical exposure,  a JEM that was used through the working life was already studied and even when the work environment have changed, application of a 4-scale at the individual level did not changed regardless of period of time considered.[2]

In Table 1, The percentage of 'Yes' for Inflammatory osteoarthritis is just 1.4%. With such a low frequency in one category, I wonder if it is appropriate to be included as a covariate in a logistic regression model

R1.3. We agree with the reviewer, the frequency of the category "Inflammatory osteoarthritis" is rather low (1.4%). However, we would prefer to do neglect this covariate for the following reasons:

  • In the population of people having inflammatory osteoarthritis, the proportion of subjects suffering from knee pain is high: 30.97% whereas, in the entire population, the proportion of subjects having knee pain is only 13.61%.
  • When testing whether the regression coefficient of the covariate "Inflammatory osteoarthritis" is null we obtain a p-value, with the Wald’s test, smaller than 10-9. Moreover, the 95% confident for the odds ratio is [2.36, 3.19]. Consequently, a subject having inflammatory osteoarthritis has a probability to have knee pain much larger than subject which does not suffer from inflammatory osteoarthritis.
  • The article by Coggon, D et al. (2000) investigate the correlation between knee pain and inflammatory osteoarthritis.[3]

We have clarified this choice in the discussion.

Line 93-95: not sure how the cut-offs 1 and 3 were determined: if Nji<=1 then Nji(b)=0 or if Nji<=3 then Nji(c)=0. There was no explanation on these two proposed cut-offs.

R1.4 We have mentioned more clearly what A, B, C models are. “ Actually, models B and C are byproducts of the classical model A, standard in the literature dealing cumulative exposure such as smoking habits or cumulative exposure index at work [1]. More precisely, In Model B, in the sum of products “duration by exposure”, we neglect small exposures by thresholding to zero exposures smaller or equal to 1.[4] In Model C, we just consider the total duration restricted to jobs physically very demanding (exposure larger than 3).[5]

Line 174 said half of sample who were overweight/obese, but Table 1 Description of the sample does not contain source data to support this conclusion.

R.1.5. Thank you for the reviewer comments, we have corrected the table, which was an old one and which was not updated with recent data.

References.

  1. Févotte J, Dananché B, Delabre L, Ducamp S, Garras L, Houot M, et al. Matgéné: a program to develop job-exposure matrices in the general population in France. Ann Occup Hyg 2011;55:865‑78.
  2. Descatha A, Despréaux T, Petit A, Bodin J, Andersen JH, Dale AM, et al. [Development of a French Job exposure matrix “MADE” for biomechanical exposure evaluation]. Santé Publique 2018;30:333‑7.
  3. Coggon D, Croft P, Kellingray S, Barrett D, McLaren M, Cooper C. Occupational physical activities and osteoarthritis of the knee. Arthritis Rheum. 2000;43:1443‑9.
  4. Fadel M, Leclerc A, Evanoff B, Dale AM, Ngabirano L, Roquelaure Y, et al. Association between occupational exposure and Dupuytren’s contracture using a job-exposure matrix and self-reported exposure in the CONSTANCES cohort. Occup Environ Med 2019;76:845‑8.
  5. Evanoff B.A. Use of a Job Exposure Matrix to Study Both Recent and Past Work Exposures on the Incidence of Carpal Tunnel Release Surgery in the CONSTANCES Cohort. [Internet]. [cité 2022 nov 24]. Available from: https://events.bizzabo.com/400073

Reviewer 2 Report

Dear authors, congratulations for the excellent work on the development of a model that allows comparing the accuracy of work duration and intensity/frequency associations in the application to severe knee pain related to musculoskeletal disorders caused by the accumulation/effect of occupational exposures during working life of a cohort of 66,553 working individuals. In addition, I reinforce the importance of the study for the innovation of the G model developed. Thus, I appreciate the research presented and the quality of the article as a whole. I would just like to hint in the discussions about the implications for potential policymakers and human resource management.

Author Response

Reviewer #2: Dear authors, congratulations for the excellent work on the development of a model that allows comparing the accuracy of work duration and intensity/frequency associations in the application to severe knee pain related to musculoskeletal disorders caused by the accumulation/effect of occupational exposures during working life of a cohort of 66,553 working individuals. In addition, I reinforce the importance of the study for the innovation of the G model developed. Thus, I appreciate the research presented and the quality of the article as a whole.  

2.1 I would just like to hint in the discussions about the implications for potential policymakers and human resource management.

R2.1 We have added as a perspective on this important point raised by the reviewer.  Implications for potential policymakers  and human resource management might been considered to achieve prevention of such pain during with the help of occupational practitioners

Reviewer 3 Report

Thank you for the opportunity to review the paper entitled "How to Use Biomechanical Job Exposure Matrices with Job History to Access Work Exposure for Musculoskeletal Disorders? Application of Mathematical Modeling in Severe Knee Pain in the Constances Cohort"

The paper is very well written and clear.

In this paper I suggest that the introduction should be more justified to focus on severe knee pain. Why knee?

What do they mean by activities at work? These can be varied and diverse. Did you take this variability into account?

It would also be pertinent to refer to the physical activity that each one has, or else be referred to as a limitation in the study.

I think the limitations are more extensive than those listed, I make the suggestion to reflect a little more on them.

You should check reference 27, I think it is incomplete.

Author Response

Reviewer #3:  Thank you for the opportunity to review the paper entitled "How to Use Biomechanical Job Exposure Matrices with Job History to Access Work Exposure for Musculoskeletal Disorders? Application of Mathematical Modeling in Severe Knee Pain in the Constances Cohort"

The paper is very well written and clear.

  1. In this paper I suggest that the introduction should be more justified to focus on severe knee pain. Why knee?

R3.1. We have chosen to focus on knee pain because it is a common disease among musculoskeletal disorders with clear work exposure. We have clarified this in the introduction.

  1. What do they mean by activities at work? These can be varied and diverse. Did you take this variability into account?

R3.2 The reviewer is right and activities at work stand for occupational mechanical exposure. Those are graduated between 0 to 4. We have mentioned this in the discussion to consider important limitations (as the two further comments).

  1. It would also be pertinent to refer to the physical activity that each one has, or else be referred to as a limitation in the study.

R3.3 We agree with the reviewer we have few information on what kind of physical activity is in the cohort. We have mentioned it as the limitation in the discussion.

  1. I think the limitations are more extensive than those listed, I make the suggestion to reflect a little more on them.

R3.4. We agree with the reviewer. As we mentioned previously, we have considered these two last comments in limitations of our study.

“Fourth, activities at work can be varied and diverse, even if the scale used can allow us to get a general idea of the exposures. Though it will never as precise as an ergonomic analysis which is however time consuming. Fifth, since leisure activities can be numerous, the mechanical exposure coming from these activities can influence musculoskeletal pain, though this degree of precision was not available in the cohort.”

  1. You should check reference 27, I think it is incomplete

R3.5 We agree with the reviewer we have corrected it